# Coastal drowning: A scoping review of burden, risk factors, and prevention strategies

**William Koon**[1,2]*, **Amy Peden**[2,3], **Jasmin C. Lawes**[1,2,4], **Robert W. Brander**[1,2]

**1** School of Biological Earth and Environmental Sciences, University of New South Wales, Sydney, New South Wales, Australia, **2** Beach Safety Research Group, University of New South Wales, Sydney, New South Wales, Australia, **3** School of Population Health, University of New South Wales, Sydney, New South Wales, Australia, **4** Surf Life Saving Australia, Sydney, New South Wales, Australia

* w.koon@unsw.edu.au

**Data Availability Statement:** All relevant data are within the paper and its Supporting Information files.

**Funding:** The authors received no specific funding for this work, however, WK's doctoral studies are

## Abstract

### Objective

Coastal drowning is a global public health problem which requires evidence to support safety initiatives. The growing multidisciplinary body of coastal drowning research and associated prevention countermeasures is diverse and has not been characterised as a whole. The objective of this scoping review was to identify key concepts, findings, evidence and research gaps in the coastal drowning literature to guide future research and inform prevention activities.

### Methods

We conducted a scoping review to identify peer reviewed studies published before May 2020 reporting either (i) fatal unintentional coastal drowning statistics from non-boating, -disaster or -occupational aetiologies; (ii) risk factors for unintentional fatal coastal drowning; or (iii) coastal drowning prevention strategies. Systematic searches were conducted in six databases, two authors independently screened studies for inclusion and one author extracted data using a standardised data charting form developed by the study team.

### Results

Of the 146 included studies, the majority (76.7%) were from high income countries, 87 (59.6%) reported coastal drowning deaths, 61 (41.8%) reported risk factors, and 88 (60.3%) reported prevention strategies. Populations, data sources and coastal water site terminology in the studies varied widely; as did reported risk factors, which most frequently related to demographics such as gender and age. Prevention strategies were commonly based on survey data or expert opinion and primarily focused on education, lifeguards and signage. Few studies (n = 10) evaluated coastal drowning prevention strategies.

### Discussion

Coastal drowning is an expansive, multidisciplinary field that demands cross-sector collaborative research. Gaps to be addressed in coastal safety research include the lack of

supported through a University International
Postgraduate Award scholarship funded by UNSW
Sydney.

**Competing interests:** The authors have declared
that no competing interests exist.

research from lower resourced settings, unclear and inconsistent terminology and reporting,
and the lack of evaluation for prevention strategies. Advancing coastal drowning science will
result in a stronger evidence base from which to design and implement effective counter-
measures that ultimately save lives and keep people safe.

## Introduction

Drowning is the process of respiratory impairment from submersion or immersion in liquid
and is considered a major global health problem [1,2]. The burden of drowning is dispropor-
tionally high in low-income countries and among males, children and young people [2,3].
Drowning outcomes include death and a range of non-fatal outcomes from survival with no
lasting consequence to survival with permanent neurological impairment [4]. The individual,
community and societal cost of drowning is immense, multi-faceted and worthy of research
that informs robust prevention efforts.

Countermeasures intended to prevent or reduce drowning are most effective when evi-
dence based. Research which delineates the nature and extent of the problem and identifies
causal factors to be addressed via intervention is therefore an a priori step in countermeasure
development [5]. For drowning, the type of body of water (e.g. swimming pool, river, ocean
etc.) is an important consideration which informs prevention efforts as the populations and
circumstances of different water sites vary [6]. In most countries, unintentional drowning
occurs more frequently at natural water sites, compared to pools or bathtubs [7].

The International Classification of Diseases' (ICD) 'natural water' category is broad and
does not specify subcategories such as lake, beach, river etc. [8]. The 2015 Utstien-style Guide-
lines for Reporting Drowning recommend 'Body of water' as a supplementary data item, pro-
viding some water type categories and the caveat that the list should be 'modified as needed to
include local hazards' [1]. While varying coding practices have limited large scale analysis of
drowning at specific water sites, these data do exist in many reporting systems and are fre-
quently available in peer reviewed and grey literature. Drowning incidence at different types of
natural water sites varies by location: in some communities, inland waterways are the primary
sites of drowning [9,10] while coastal waters represent a major hazard in others. For example,
948 drowning deaths, 85% of all drowning cases, occurred in the sea over a five-year period in
a coastal province in Iran [11]; 399 fatal and non-fatal drowning events, 90% of drowning all
incidents, occurred at the coast across a 10-year period in Marseille, France [12]; and 78
drowning deaths, 76% of all drowning fatalities, occurred in the ocean in Port Elizabeth, South
Africa [13].

Defining coastal bodies of water for the purposes of drowning data management and pre-
vention activities has taken different forms [14,15]. While no international consensus list of
coastal drowning water sites exists, an operationalised definition for coastal waters might
include beaches, harbors, bays, rock platforms, estuaries, wetlands, lagoons, saltwater marshes,
natural occurring saltwater canals and intercoastal waterways, large inland seas such as the
Black Sea or the Caspian Sea, the Great Lakes in North America and the open ocean itself.
These diverse environments present challenges to coastal drowning research and prevention
efforts: coastal hazards are variant and complex [16–18], their study involves many different
scientific disciplines and subsequent control measures are the domain of multiple sectors.
Maritime safety likely represents the most robust drowning prevention initiatives; centuries of
seafaring and large-scale public and private investment have resulted in a mature field of safety

science which supports all manners of ergonomic, education and enforceable policy counter-measures [19]. While arguably not as advanced as its naval counterpart, nearshore coastal drowning prevention and safety also has a rich history and rapidly developing scientific basis [20–22].

Early coastal safety research frequently took the form of epidemiologic studies: descriptive case series from the 1960's identified the importance of beach rescuers [23], difficulty in mea-suring exposure [24], and the dangers of certain activities such as scuba diving [25]. In the 1980s, seminal work by Wright and Short typified geomorphological beach states which later became the basis for the Australian Beach Safety and Management Program (ABSAMP) and other similar risk management systems used around the world [26–29]. Studies on the rip cur-rent hazard have evolved from observations of sticks floating out to sea in the 1940s to complex experiments involving Global Positioning Satellite (GPS) drifters and, recently, human dimen-sions involving identification and escape strategies [16,30–32]. The past decade has seen an increase in coastal drowning and safety studies employing multidisciplinary and social science methods and advances in specific areas such as beach safety [22], rock platforms and rock fish-ing [33], and surf lifeguarding [34].

While obviously expansive, the extent, range and nature of coastal drowning research has not been characterised as a whole. Scoping reviews are useful for this purpose; they aim to identify key concepts, findings, evidence and research gaps of a particular field of study [35]. Scoping reviews are conducted systematically with rigorous and transparent methods but dif-fer from traditional systematic review or meta-analysis methods [35]. In a scoping review, the primary interest is the identification and mapping of characteristics and concepts in a field of work versus the evaluation of a clinically meaningful question to inform practice [36]. The breadth of coastal drowning research, scale of coastal drowning prevention efforts and the absence of any existing effort to this effect justify a broad assessment of this nature.

The primary aim of this review was to systematically characterise peer-reviewed coastal drowning research to better understand the science driving associated safety initiatives. Specif-ically, we sought to identify key concepts and factors studied, describing how they have been analysed and discussed, in order to synthesize present understanding and highlight gaps in the field. The review focused on fatal unintentional coastal drowning from non-boating, -disaster or -occupational origin, and employed an analytic framework based on coastal drowning study characteristics, epidemiological burden, risk factors and prevention strategies. Under-pinning this study is the belief that a critical examination of the state of coastal drowning sci-ence will clarify current understanding of the field and help plot a course for future research and prevention activities which seek to save lives.

## Materials and methods

We identified and conducted a scoping review of the published coastal drowning literature using the five-stage Arksey and O'Malley framework [35], with incorporated recommenda-tions from Levec et al. [37]. We also utilised a modified Preferred Reporting Items for the Sys-tematic Reviews and Meta-Analyses (PRISMA) methodology to guide the review (S1 File) [38,39].

### Stage one: Identifying and refining research questions

For a manageable review of such a broad topic, we refined the scope of inquiry to focus on fatal unintentional coastal drowning, excluding boating, disaster and occupational aetiologies. While many drowning prevention countermeasures are universal, those involving intentional drowning or submersion from boating, disaster (e.g. flood, tsunami, hurricane) or

occupational situations involve some specific mechanisms. Many of the key concepts and evidence from research in these drowning sub-fields are distinct, and several systematic reviews exist specific to these subjects [40–44]. This refined focus was further guided by three domains consistent with a public health approach: epidemiological burden, risk factors and primary prevention strategies [45].

## Stage two: Identifying relevant studies

The multidisciplinary nature of coastal drowning as a field of research means that relevant studies come from a diverse array of journals. We searched MEDLINE, MEDLINE epub, EMBASE, Environment Complete, CINAHL and SPORTDiscus databases for peer reviewed articles published in English or Spanish up to 30 April 2020. With the aid of a specialist librarian, these databases were selected strategically because they index journals from a variety of disciplines which have previously published relevant coastal drowning studies.

The complete search strategies for each database are available in S2 File but generally followed a format of 'drown* AND' keywords related to coastal water sites (e.g. 'ocean', 'sea', 'beach', 'harbor', 'coast' etc.), limited to human studies. Notably, through the iterative process of developing database search strategies we discovered that multiple relevant studies from health sciences journals were not appearing in our searches based solely on coastal water site terms. Many studies reported and discussed coastal drowning in the text, but primarily focused on swimming pool drowning and therefore only included that information in the title, abstract and searchable metadata. We therefore opted to include 'swimming pool*' in our search of health science databases to ensure we captured all potential studies. We also manually checked references of included studies to identify additional papers which met the inclusion criteria, and cross referenced our own databases of relevant studies to ensure maximum inclusion.

## Stage three: Study selection

Study inclusion criteria followed an iterative development process as recommended by Levac et al. [37]. Studies included in this scoping review must have: (a) been written in English or Spanish, (b) been an original peer-reviewed article presenting new data (e.g. not a review, editorial, or conference abstract); and either (c) reported 10 or more fatal unintentional coastal drowning cases which did not arise from boating/transport, disaster, or occupational origin; (d) reported risk factors for unintentional fatal coastal drowning; and/or (e) reported prevention strategies consistent with the "Preparation" or "Prevention" phases of the 2016 drowning timeline (Fig 1) [46]. For criteria (a), author WK is fluent in Spanish and screened those studies. For criteria (c), reported drowning statistics, we excluded studies which only reported drowning cases from 'natural' or 'open' bodies of water if we could not determine that those water sites were coastal, and those reporting less than 10 coastal drowning cases.

Two authors (WK and AP) independently screened titles and abstracts using the online software Covidence and remained in contact throughout the process as recommended by Levac et al. to discuss disagreements, challenges and uncertainties [46,47]. WK and AP reviewed full texts of remaining papers separately, (82% agreement, Cohen's Kappa = 0.64) and discussed disagreements until reaching consensus. A third reviewer (RB) was available to determine final inclusion but was not required.

## Stage four: Charting the data

One author (WK) completed data extraction using a standardised form developed by the study team. The form (S3 File) involved four sections: study information (country and year of publication, aims etc.), epidemiological burden, risk factors and prevention strategies. The form

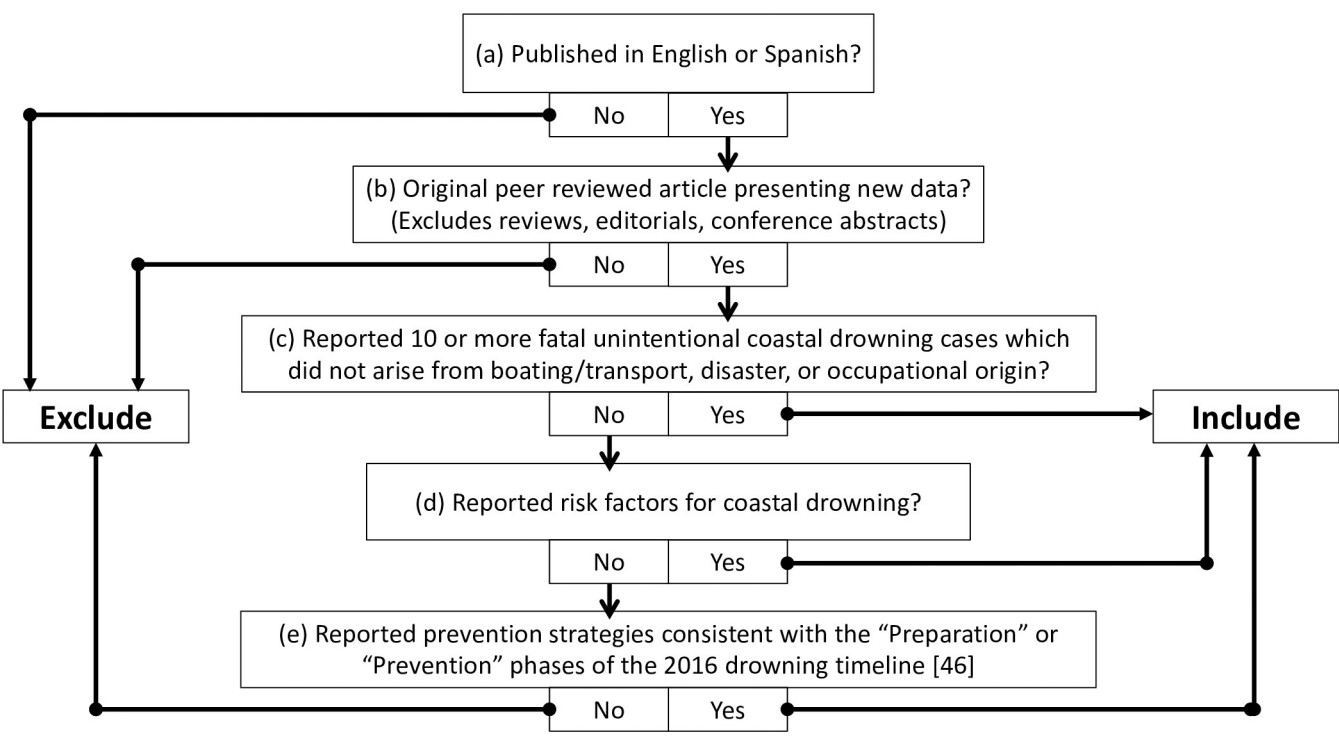

**Fig 1. Study inclusion-exclusion decision tree flow chart.**

was piloted on five randomly chosen studies to ensure usability and accurate data capture [37]. The development and pilot process identified the need for a series of rules to ensure consistency, detailed below.

**Stage 4.1: Charting epidemiological data.** Among other information, we endeavoured to record the number of reported coastal drowning deaths reported in each study and the terminology used to identify those cases as 'coastal'. We only noted the primary data on coastal drowning deaths reported and analysed each study, generally in the results section, and excluded information referenced, but not analysed, for example information reported in an introduction or background section. If a study reported both fatal and non-fatal unintentional drowning in the coastal environment, we extracted only data related to drowning cases which we could confirm as fatal. We noted if studies reported intentional or boating cases separately, extracting only data relating to the unintentional/non-boating cases where possible and noting all coastal drowning deaths when not specifically separated. Where two or more papers used the same dataset for the exact same timeframe, we only extracted the reported cases one time from the paper providing the most information (e.g. a paper reporting cases and risk factors over a paper only reporting cases).

**Stage 4.2: Charting data on risk factors.** To better understand coastal drowning epistemology, we purposefully adopted a broad approach in charting associated risk factors. Drowning is a process with several phases before death [46], we therefore recorded information on indicators that may be associated with fatal outcome and those pertaining to other preceding/proximal outcomes, such as those related to knowledge, attitudes, ability etc. which could increase coastal drowning risk [48]. For a comprehensive perspective, hereafter we refer to indicators, whether in reference to fatal or preceding/proximal outcomes, proven causal association or not, as risk factors. We noted the risk factor (e.g. male, young age, weekend etc.), the outcome associated with that factor (e.g. fatal coastal drowning, low swimming skills, unsafe

swim choice, etc.) and the method for determining the risk factor (e.g. statistical analysis of epidemiological data, descriptive analysis of survey data, etc.).

For epidemiological studies reporting counts of fatal coastal drowning by some category, we noted the group with the most deaths as a risk factor and the method determining the risk factor as 'higher proportion of cases'. For example, in a study presenting the number of coastal drowning deaths by age group, the age group with the most deaths would be noted as a risk factor determined by higher proportion of cases. Additionally, we only charted risk factors based on epidemiological data (fatal drowning cases) if they specifically related to coastal water sites. We did not include those factors based on data from, or intended for, non-coastal or multiple bodies of water where information for the coastal environment couldn't be isolated. Although many risk factors are likely to be ubiquitous, we were interested in identifying those factors which have been characterised in the literature as specific to coastal drowning.

**Stage 4.3: Charting data on prevention strategies.** For those studies that reported prevention strategies, we extracted information on the recommendation/strategy and the data or evidence on which it was made, if any. Prevention strategy recommendations made without supporting data were classified as 'expert opinion'. Each prevention recommendation was categorized based on its focus, such as education, lifeguards, signage etc. for summary. As the primary aim of this scoping review was to identify and characterise key concepts and factors in the coastal drowning literature, and not provide a critically appraised synthesised result for a specific clinically meaningful question, we did not perform methodological limitations or risk of bias assessments [36].

### Stage five: Presenting the data

Charted data were summarised via numerical summary and thematic analysis [35,37], and reported as single cohort of studies and separately by the domains described in stage one: epidemiological data, risk factors or prevention strategies. For presenting data on risk factors, we described the number of times factors were identified in the papers as 'risk factor reports.' Summarising the data in this way allowed us to map not only the unique factors identified in the literature, but also the number of times they have been studied, have used different analyses and employed different outcome measures.

## Results

Database searches identified 1,167 unique studies, 110 remained after screening and full-text review. An additional 36 relevant studies were identified by hand-searching references, resulting in 146 included studies (Fig 2; S1 Table). Of these, 22 (15.1%) reported fatal coastal drowning statistics, risk factors and prevention strategies, while the remaining studies reported one or a combination of the three domains. Although a small number of studies published in Spanish were reviewed in the screening stage, none met inclusion criteria resulting in a final cohort of only English language studies.

### Study characteristics

More than half the studies (52.7%) came from Australia and the United States (49 [33.6%] and 28 [19.2%] respectively), 13 studies (8.9%) were from New Zealand and five (3.4%) each were from Iran, South Africa and the United Kingdom. Five studies (3.4%) included data from more than one country, and the remaining 36 studies (24.7%) came from 19 countries (Table 1). Classified according to World Bank Country Income and Lending Groups at the time of publication (or nearest year from available data), 112 studies (76.7%) were from high income countries, 21 studies (14.4%) came from upper middle-income countries, 9 studies

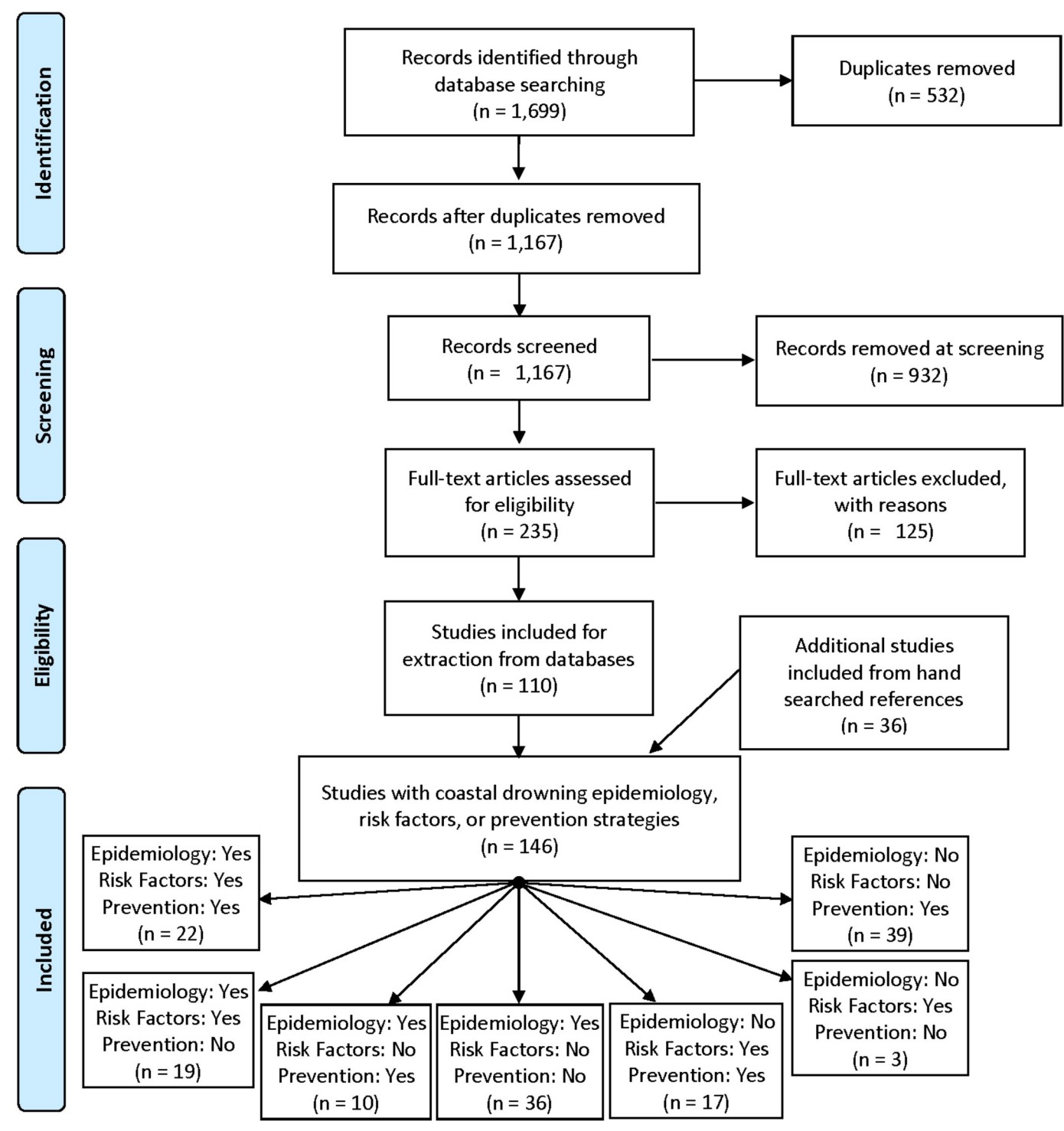

**Fig 2. Modified PRISMA-ScR flow chart.**

(6.2%) were from lower middle-income countries, and no studies were from low-income countries [49]. Of the five multi-country studies, two were global surveys with responses from multiple countries and three only included data from high income nations. The earliest study

**Table 1. Number of studies based on country of included data.**

| Country | Number of Studies | Percent of total |
|---|---|---|
| Australia | 49 | 33.6% |
| USA | 28 | 19.2% |
| New Zealand | 13 | 8.9% |
| Iran | 5 | 3.4% |
| Multiple | 5 | 3.4% |
| South Africa | 5 | 3.4% |
| UK | 5 | 3.4% |
| Brazil | 4 | 2.7% |
| Spain | 4 | 2.7% |
| France | 3 | 2.1% |
| India | 3 | 2.1% |
| Turkey | 3 | 2.1% |
| China | 2 | 1.4% |
| Costa Rica | 2 | 1.4% |
| Korea | 2 | 1.4% |
| Pakistan | 2 | 1.4% |
| Singapore | 2 | 1.4% |
| Fiji | 1 | 0.7% |
| Finland | 1 | 0.7% |
| Ghana | 1 | 0.7% |
| Israel | 1 | 0.7% |
| Mexico | 1 | 0.7% |
| Norway | 1 | 0.7% |
| Scotland | 1 | 0.7% |
| Thailand | 1 | 0.7% |
| United Arab Emirates | 1 | 0.7% |

included in this review was published in 1963 from the United Kingdom [23]. Since then, studies reporting coastal drowning statistics, risk factors and prevention strategies have steadily increased to date (Fig 3). This cohort of coastal drowning research was published in 66 different journals; the three most frequently represented disciplines were non-injury-specific health/medical sciences (62 studies, 42.5%), followed by the physical sciences (32 studies, 21.9%), and those which were injury- or safety-specific (18 studies, 12.3%; Table 2).

## Epidemiological burden

Of the 146 included studies, 87 (59.6%) reported a combined 21,095 coastal drowning deaths using various terminology to indicate coastal waters (Table 3). About a quarter of these papers (n = 21, 24.1%) grouped terminology from more than one body of water into a single category (e.g. "Ocean and Bay"). Most studies (n = 57, 65.5%) included multiple types of bodies of water, including non-coastal sites. In these multi-water site studies, coastal drowning typically represented between 10% and 40% of drowning deaths reported; coastal waters represented the category with the most deaths in half the studies (n = 29, 50.9%).

**Case inclusion and data sources.** The geographic catchment for case inclusion varied with many studies (n = 36, 41.4%) reporting data at the country level, a quarter (n = 22, 25.3%) at the state or provincial level, one fifth (n = 17, 19.5%) at the regional or sub-state level, nine (10.3%) at the local government (city/town/municipal) level, and three (3.4%) at the sub-

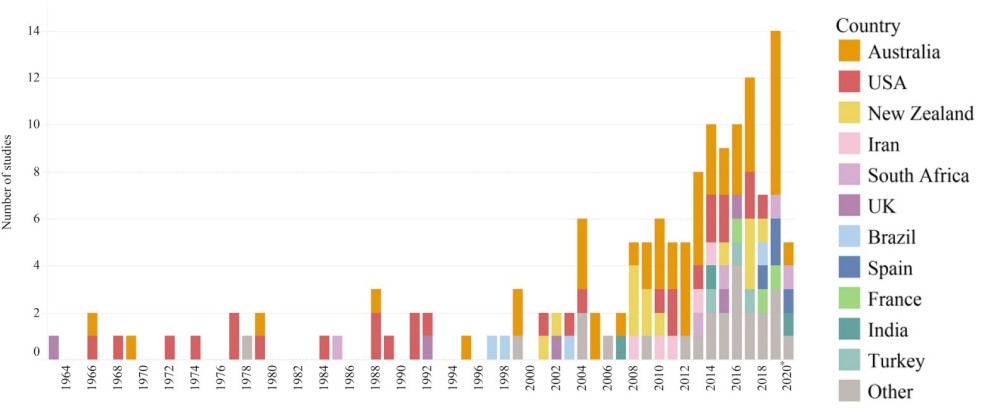

**Fig 3. Studies by publication year.** *Note, 2020 current through April 30.

municipal level. Five of the state/province studies included multiple states or provinces. Of the country level studies, two included multiple countries and, separately, two were partially complete: both reported national data excluding a single state.

Most studies (n = 67, 77.0%) included all ages of drowning deaths, while 11 (12.6%) reported only adults and nine (10.3%) included only children and/or adolescents (0–19 years). The majority of studies (n = 80, 92.0%) included only drowning cases while seven (8.0%) included drowning and other causes of death. Many studies (n = 34, 39.1%) only included cases from a specific cohort, such as drowning deaths from the beach (n = 10, 11.5%), rip currents (n = 7, 8.0%), SCUBA/skin diving or snorkelling (n = 5, 5.7%), or incidents involving a lifeguard or lifesaver (n = 5, 5.7%). The majority of studies (n = 50, 57.5%) used multiple data sources to report coastal drowning statistics (Table 4).

**Reporting of intentional, boating, and occupational drowning.** Studies which focused solely on intentional, boating, disaster or occupational drowning were excluded at the screening stage, but several of the included studies contained various facets of these sub-topics. Occupational drowning cases were included or discussed in six papers, but none reported work-related deaths by body of water, thus the number of non-occupational coastal drowning deaths

**Table 2. Peer-reviewed academic journals with three or more included studies.**

| Journal | Count of studies (%) | Journal Discipline |
|---|---|---|
| International Journal of Aquatic Research & Education | 13 (8.9%) | Industry specific |
| Natural Hazards | 13 (8.9%) | Physical sciences |
| Ocean & Coastal Management | 9 (6.2%) | Physical sciences |
| International Journal of Injury Control & Safety Promotion | 7 (4.8%) | Injury/safety |
| Journal of Coastal Research | 7 (4.8%) | Physical sciences |
| Medical Journal of Australia | 7 (4.8%) | Other health/medical sciences |
| Accident Analysis & Prevention | 5 (3.4%) | Injury/safety |
| Australian & New Zealand Journal of Public Health | 4 (2.7%) | Other health/medical sciences |
| Health Promotion Journal of Australia | 3 (2.1%) | Other health/medical sciences |
| Natural Hazards & Earth System Sciences | 3 (2.1%) | Physical sciences |
| PLOS ONE | 3 (2.1%) | Interdisciplinary |
| Public Health Reports | 3 (2.1%) | Other health/medical sciences |
| Resuscitation | 3 (2.1%) | Other health/medical sciences |
| Tourism Management | 3 (2.1%) | Industry specific |

**Table 3. Number of studies by coastal terminology used in drowning death body of water categories.**

| Coastal water category terminology | Number of studies (N = 87) |
|---|---|
| Ocean | 29 (33.3%) |
| Sea | 23 (26.4%) |
| Beach | 21 (24.1%) |
| Bay | 11 (12.6%) |
| Not specific [a] | 10 (11.5%) |
| Surf | 9 (10.3%) |
| Other [b] | 9 (10.3%) |
| Saltwater | 8 (9.2%) |
| Harbour | 8 (9.2%) |
| Region specific body of water | 6 (6.9%) |
| Coast | 3 (3.4%) |
| Rocks | 3 (3.4%) |
| Lagoon | 2 (2.3%) |
| Canal | 2 (2.3%) |

[a] Confirmed coastal only papers, no specific terminology for body of water used.

[b] Other terminology includes saltwater ocean pool, estuary, tidal creek, saltwater inlet, saltwater sound, saltwater jetty, shoreline, lake surf (North American Great Lakes), and marine environment.

from these studies is unknown. The majority of studies did not clearly differentiate between boating and non-boating cases: 32 studies (36.8%) did not include or address boating in any way and 23 studies (26.4%) included or discussed boating in some manner, but did not report boating and non-boating cases separately. In 14 studies (16.1%), boating cases were not discussed, but were unlikely to be present as they focused on specific cases such as rip current or SCUBA drowning. Eleven studies (12.6%) reported coastal boating and non-boating drowning separately, and seven (8.0%) explicitly included only non-boating cases. While most studies (n = 36, 41.4%) included only unintentional cases, more than one third (n = 32, 36.8%) did not address or discuss intentionality in any way. Eleven studies (12.6%) included, but did not report intentional cases separately by water site, five (5.7%) included and did report intentional cases by water site and three (3.4%) did not discuss but were unlikely to include intentional cases based on the study focus (e.g. bystander rescuer, SCUBA).

## Risk factors

One hundred seven unique factors were identified from a total of 228 risk factor reports in 61 of the included studies (41.8%). The majority of risk factor reports (n = 120, 52.6%) used epidemiological data; 84 (36.6%) used survey data, 20 (8.8%) used expert panel techniques and four (1.8%) was a qualitative study.

**Outcomes and analysis.** Fatal coastal drowning was the primary outcome in the majority (n = 137, 60.1%) of risk factor reports: 120 from epidemiological data and 17 from a single study which employed an expert panel technique to determine factors that influence a user's risk of drowning at surf beaches [50]. Fatal coastal drowning risk factor reports using epidemiological data (n = 120) were most frequently identified by a higher proportion of cases (n = 98, 43.0%), followed by statistical analysis with a significant result (n = 14, 6.1%) and a reported incidence or prevalence rate (n = 8, 3.5%).

Preceding/proximal outcomes were used in 91 (39.9%) risk factor reports, and related to knowledge (n = 22, 9.6%), skills or abilities (n = 19, 8.3%), behaviour or practice (n = 18, 7.9%)

**Table 4. Data sources from studies which reported coastal drowning statistics.**

| Data Source | Single sourced studies (n = 37) | Studies with multiple data sources (n = 50) | Total (n = 87) |
|---|---|---|---|
| Death certificate or coroner data | 17 | 38 | 55 (63.2%) |
| Media reports | 6 | 21 | 27 (31.0%) |
| Police records | 2 | 23 | 25 (28.7%) |
| Hospital data | - | 20 | 20 (23.0%) |
| Lifeguard or beach rescue data | 3 | 10 | 13 (14.9%) |
| Emergency Medical Services | 1 | 9 | 10 (11.5%) |
| Other [a] | 1 | 11 | 12 (13.8%) |
| Questionnaire | 1 | 6 | 7 (8.0%) |
| Unclear | 4 | - | 4 (4.6%) |
| Other medical data source (i.e. Clinic, Private Practice, Drowning Centres) | - | 4 | 4 (4.6%) |
| Other rescue service (e.g. fire department) | 2 | 2 | 4 (4.6%) |

[a] Other data sources include various government agencies (e.g. Ministry of the Interior, Ministry of Health), industry equipment malfunction reports (generally for SCUBA studies), and non-profit organizations.

decision-making (n = 17, 7.5%), and awareness or risk perception (n = 15, 6.6%). Of these preceding/proximal outcome risk factors (n = 91), the majority (n = 65, 71.4%) were identified by statistical analysis of survey data, 19 (20.9%) were identified by descriptive survey data, four (4.4%) were identified via qualitative methods, and three (3.3%) were identified through expert panel techniques (Table 5).

**Factors studied.** Most of the 228 risk factor reports related to the person (n = 180, 78.9%), followed by the environment (n = 26, 11.4) and time (n = 22, 9.6%). Reported risk factors most frequently related to gender, behaviour/activity, age, and race/ethnicity, and the single most frequently reported risk factor was being male (Table 6, S2 Table).

## Prevention strategies

Over half of the included studies (n = 88, 60.3%) recommended coastal drowning countermeasures. A principal prevention focus was education, recommended in 70 studies (79.5%), followed by various strategies related to lifeguards and signage, discussed in 24 (27.3%) and 19 (21.6%) studies, respectively. Prevention recommendations were most frequently supported by survey analysis (31 studies, 35.2%), followed by expert opinion without explicit supporting data (24 studies, 27.3%) and epidemiological information (21 studies, 23.9%). Ten studies (11.4%) made recommendations based on analysis which aimed to evaluate a specific strategy. Table 7 shows how many papers recommended strategies within each prevention category by the evidence base presented to make the recommendation.

The 70 studies which discussed education strategies made varied recommendations. Nearly half (n = 32, 45.7%) suggested the promotion of specific educational messages, most frequently related to swimming in lifeguard supervised areas (e.g. "swim between the flags", "swim in front of a lifeguard"), rip currents (identification of, and escape/survival) and personal ability (e.g. "know your limits"). Rip current education was discussed in 24 studies (34.3%), 16 (22.9%) recommended campaigns or the use of mass media, 13 (18.6%) discussed lifeguard supervised zones or flagged patrol areas, and seven (10%) each recommended school-based programs and teaching skills such as resuscitation or basic rescue techniques. About one third of the educational strategy studies (n = 31.4%) included recommendations on *how* to conduct

**Table 5. Number of risk factor reports by preceding/proximal outcomes (not fatal drowning) and analysis used to identify the risk factor.**

| Outcome Category | Preceding/Proximal Risk Factor Outcome | Description of survey data | Statistical analysis of survey data | Qualitative | Expert Panel | Total |
|---|---|---|---|---|---|---|
| Awareness or perception of risk | | | | | | |
| | Low awareness or perception of risk, threat, vulnerability or response efficacy | 2 | 9 | 1 | - | 12 |
| | Overestimation of/overconfidence in ability to cope with risk | - | 3 | - | - | 3 |
| Behaviour or practice | | | | | | |
| | More likely to have difficulty in the water | | | - | 3 | 3 |
| | Drink alcohol while fishing | - | 2 | - | - | 2 |
| | Less likely to see beach safety flags or signs | - | 1 | - | - | 1 |
| | Less likely to check weather before fishing | - | 2 | - | - | 2 |
| | Less likely to heed warning from flags/sign/lifeguard | 4 | 1 | 1 | - | 6 |
| | Less likely to signal for help | 1 | - | - | - | 1 |
| | Swims at the beach more (increased exposure) | - | 3 | - | - | 3 |
| Decision-making or knowledge | | | | - | - | |
| | Less likely to make safe swim choice or decision | 1 | 16 | - | - | 17 |
| Knowledge | | | | | | |
| | Low rip/ocean knowledge | 9 | 5 | - | - | 14 |
| | Low safety knowledge | - | 6 | 1 | - | 7 |
| | Lower self-efficacy in local knowledge and sea experience | - | 1 | - | - | 1 |
| Skill or ability | | | | | | |
| | Less likely to correctly identify hazards | - | 5 | - | - | 5 |
| | Low swim ability (self-reported) | - | 6 | 1 | - | 7 |
| | Lower surf swim confidence | - | 3 | - | - | 3 |
| | Lower swimming ability confidence | 2 | - | - | - | 2 |
| | No CPR Skills | - | 2 | - | - | 2 |
| **Total** | | **19** | **65** | **4** | **3** | **91** |

these activities; the most common related to working with specific priority populations (n = 6, 8.6%) and elements related to cultural or linguistic appropriateness (n = 5, 7.1%).

Of the prevention strategy studies, relatively few (n = 10, 11.36%) involved evaluation. Hatfield et al. [51] and Houser et al. [52] both evaluated rip current information campaigns, Davoudi-Kiakalayeh [53] and Moran [54] evaluated multi-year safety programs with several different components, and Wilks et al. [55] and Barcala-Fuerlos et al. [56] both evaluated beach safety education programs for children. Matthews et al. [57] tested the effect of different variations of beach signage, Warton et al. [58] conducted a global survey exploring the effect of a reality-based television show (*Bondi Rescue)* on beach safety attitudes and knowledge, and McCarroll et al. [32] and Van Leeuwen et al. [59] conducted human trial experiments to determine which rip current escape method was most effective and thus worthy of promotion.

## Discussion

This scoping review of 146 coastal drowning studies provides insight into the collective understanding of the coastal drowning problem and the science underpinning efforts to progress safety in this area. Our goal was to identify key concepts and factors in the peer reviewed coastal drowning literature and while some points of synthesis were identified, this review primarily shows the vast heterogeneity of science in the field. The factors and topics studied, how they have been investigated and what new knowledge they contributed are wide ranging and

**Table 6. Number of risk factor reports by risk factor in each group by outcome and analysis used.**

| Risk Factor Group | Fatal Coastal Drowning Outcome | | | | Other Preceding/proximal outcome | | | | Total |
|---|---|---|---|---|---|---|---|---|---|
| | Higher proportion of cases | Statistical analysis of epidemiological data | Expert panel/judge technique | Higher incidence or prevalence rate | Expert panel/judge technique | Description of survey data | Qualitative methods | Statistical analysis of survey data | |
| Gender | 21 | 1 | | 1 | | 1 | | 15 | 39 |
| Behavior/activity | 14 | 1 | 8 | | | 4 | | 3 | 30 |
| Age | 14 | 1 | | 4 | | 4 | | 6 | 29 |
| Residence | 9 | | | 1 | | 2 | 4 | 11 | 27 |
| Race/ethnicity | 2 | | | 1 | | 3 | | 19 | 25 |
| Time (Hour, Day, Season) | 22 | | | | | | | | 22 |
| Other Demographic or Person Factor | 4 | 3 | 1 | 1 | | 3 | | 5 | 17 |
| Beach Characteristics (physical/social) or water conditions | 5 | 4 | 4 | | 1 | | | | 14 |
| Knowledge, belief, level of experience | | | 2 | | 1 | 1 | | 4 | 8 |
| Weather | 4 | 4 | | | | | | | 8 |
| Skill | 1 | | | | 1 | 1 | | 2 | 5 |
| Dangerous condition warnings or signage | 2 | | 2 | | | | | | 4 |
| Total | 98 | 14 | 17 | 8 | 3 | 19 | 4 | 65 | 228 |

diverse. The populations included, data sources employed, and terminology used varied widely. A myriad of risk factors for coastal drowning and other preceding/proximal outcomes have been identified through analysis of different types of data using various methods. Recommended prevention strategies most frequently related to education, were commonly based on survey data or expert opinion alone and were rarely evaluated. Of note, the field's scientific agenda to date has been, and continues to be, dominated by research from a small number of high-income countries.

**Table 7. Number of papers by prevention strategy and supporting evidence.**

| Evidence base for recommendation | Prevention Strategy Group | | | | |
|---|---|---|---|---|---|
| | Education | Lifeguards | Signage | Other [a] | Grand Total [b] |
| Survey Data | 28 | 5 | 6 | 3 | 31 |
| Expert opinion—no supporting data presented | 14 | 7 | 6 | 4 | 24 |
| Epidemiological information | 11 | 10 | 1 | 6 | 21 |
| Specific evaluation of strategy | 9 | 1 | 2 | | 10 |
| Other data [c] | 8 | 1 | 1 | 2 | 8 |
| Data from physical science methods (GIS, Geomorphological beach survey, Rip current drifter experiment etc.) | 4 | 1 | 3 | 1 | 5 |
| Grand Total | 70 | 24 | 19 | 16 | 88 |

[a] Other prevention strategies related to designated swim areas (n = 6), public rescue equipment (n = 4), guidelines or policy (n = 3), lifejackets (n = 3), technology (n = 2), design and implementation of a beach safety management program (n = 1), a call for collaboration around beach safety efforts (n = 1), and a recommendation to prioritize a certain population with beach safety interventions (n = 1).

[b] Totals may be more than 100% as some papers made more than one recommendation per category, based on different data.

[c] Other data includes beach usage information (n = 3), qualitative data (n = 2), image analysis (n = 1), video content analysis (n = 1) and expert panel techniques (n = 1).

## Coastal drowning as a field of research

The multidisciplinary nature of coastal drowning prevention as a field of practice and research is undoubtedly beneficial, but made this review particularly difficult. Lack of consistent terminology and absence of coastal-specific searchable metadata in databases made identifying studies in the early phases of this review challenging: a quarter of the final included articles were found by hand searching references from database identified papers. Moreover, coastal drowning studies are published in a wide array of journals, which might limit exposure of important work across disciplines. Although frequently framed as an injury prevention or public health issue [60], those attempting to find coastal drowning research must look outside these fields, and even beyond the health sciences for relevant studies. As there appears to be no distinct 'home' for coastal safety research, breaking down silos and facilitating interdisciplinary and cross sector collaborative research is imperative for future progress in the field.

The most striking research gap identified by this review was the paucity of studies on coastal drowning from non-high-income settings, where the burden of drowning is much greater [2,3]. Although we do not know the proportion of coastal water drowning deaths from many lower resource locations, robust estimation models from the Global Burden of Disease (GBD) study associated "landlocked" nations with decreased drowning mortality, meaning countries with coastlines had systematically higher estimated drowning mortality [3]. These lower-income coastal communities, not the high-income nations which have traditionally dominated coastal safety research, are most likely to benefit from advances in coastal drowning prevention science and practice.

## Epidemiological burden

Our attempt to chart data from the published literature on the epidemiological burden of fatal unintentional non-boating non-occupational coastal drowning proved to be problematic, but in itself identified a gap and provided important learnings. Only 20 studies, less than a quarter of those which reported fatal drowning cases, allowed us to extract unintentional non-boating cases, and none reported occupational drowning separately by body of water. While a limitation of this review's original intent, the effort highlighted the importance of collecting and reporting these types of data. Information from the "Precipitating event" core category of the Utstien-Style Guidelines for Reporting Drowning, which would include intentionality, boating, disaster etc., is imperative to establishing factors in the drowning process which can be prioritized for intervention [1]. Without these data, advancing research to actionable prevention schemes becomes difficult.

Inclusion of these and other types of data may have been a function of data sources used and/or populations studied, which are frequently determinants of one another. Coroner and death certificate data were used in the majority of studies, and in many it was the only data source. While vital statistics records in some regions offer robust research opportunities, others require linkage with other systems to provide rich enough datasets for meaningful analysis. Of note, data systems which gather information from the scene of the drowning such as police, lifeguard, fire department or other emergency service reports provide important data not found in clinical or death records. Whenever possible, scene-based data should be prioritized in tandem with forensic information to establish linked datasets which offer opportunities to study factors from across the drowning process.

Terminology varied in the identification of coastal bodies of water as locations for drowning, inhibiting meaningful conclusions on the relative burden to other types of bodies of water. While various categories of coastal water sites represented the most deaths in half of the studies that reported multiple bodies of water, different and inconsistent use of terms limit the

usefulness of such a finding. The water site category with the most deaths depends entirely on the vocabulary used to report results. For example, a study describing drowning deaths in terms of 'saltwater' and 'freshwater' might report an overwhelming number of 'freshwater' cases, but the same study using water site categories of 'ocean', 'lake', 'swimming pool' and 'bathtub' may show 'ocean' to have the most deaths as 'freshwater' cases are spread into other groupings. A standardized list of terms with definitions for different types of bodies of water may seem to be a logical path toward improvement, but such a list is likely to be cumbersome and has the potential to exclude local context important for countermeasure development. Future movement towards such a list should carefully consider its usefulness and potential limits.

## Risk factors

Research identifying risk factors related to fatal and other relevant preceding/proximal outcomes is a cornerstone of the coastal drowning field. The plethora of factors studied, many multiple times with different outcome measures, is a strength of this body of work. Reviewing factors associated with preceding/proximal non-death outcomes alongside those associated with fatal results offers a richer, more contextualised understanding and could be helpful for designing countermeasures. Conversely, nearly 43% of the risk factor reports were based only on a greater proportion of cases, less a reflection of increased likelihood of outcome and more so of burden influenced and potentially skewed by varying populations, sample selection, exposure or any number of other confounders. Only six papers, four published pre-1990, identified risk factors based on higher incidence or prevalence rates. Various complications with presenting population-based drowning rates have been identified [61], and measuring exposure has and continues to be a major challenge [24,62]. Future studies seeking to identify coastal drowning risk factors should strive to present both case counts and incidence or prevalence rates as both are important for prioritized interventions.

Limitations notwithstanding, several important risk factors were identified in this review as relevant for coastal drowning, some of which have also been recognized as risk factors for other bodies of water [63]. The clearest of these is being male, a group identified for being at particular drowning risk in several specific studies [64–66], and in other systematic drowning reviews [10,67]. Age was also highlighted as a risk factor for coastal drowning by several studies. Although clear consensus was lacking (age groups used were not consistent), most studies that discussed age identified the young adult and teenage years as life periods of higher risk (S2 Table). Other frequently cited person-based risk factors included race and ethnicity, mostly identified by statistical analysis of survey data; and residency related factors such as being a tourist, living near the coast, or being born in another country. Where coastal drowning specific risk factors align with universal drowning risk factors (i.e. being male), collaboration on countermeasure strategies among drowning prevention organisations is vital.

Only nine papers included risk factor reports (n = 22) related to physical or social beach characteristics, water conditions, or weather. A hallmark of multidisciplinary coastal safety research is the intersection of the physical science, namely oceanography and coastal geomorphology, and disciplines such as public health and psychology relating to the human beings who interact with the coastal environment. Yet, surprisingly, few studies have correlated water or other environmental conditions to increased likelihood of fatal drowning. Some studies have investigated lifeguard rescues with ocean conditions [68–70], but several questions remain as to the nature of the causal relationship between these factors and coastal drowning. Future research characterizing these associations, and the interaction between environmental and person-based factors, would be of great benefit to those involved in prevention efforts.

## Prevention strategies

The majority of studies discussed prevention strategies specific to coastal drowning. Collectively, the recommendations were vague, based on limited evidence or expert opinion and primarily related to three themes: education, lifeguards, and signage. Moreover, evaluation of coastal drowning prevention efforts is largely missing from the field, thus providing little guidance to those tasked with funding and implementing such strategies.

Education was the most frequently discussed coastal drowning prevention strategy in this cohort of peer reviewed literature, and along with the provision of lifeguard supervision, likely represents the principle coastal drowning prevention activity worldwide [71]. Although an important component, it is important to note that increased knowledge or awareness alone does not necessarily lead to changes in safety behaviour [72]. Some studies incorporated health theory and/or behaviour change approaches [51,73–79], which are likely to make education efforts more efficacious [80], but many studies provided only ambiguous suggestions for educating the public or increasing awareness. Education efforts that seek to improve knowledge, increase awareness of risk and inform attitudes and beliefs play a central role in a cohesive systems level approach to preventing coastal drowning, however, future research must consider return on investment and if viable, the optimal messages, methods and age at which this education should be delivered.

Recommendations related to lifeguards (e.g. establishing or expanding services, swimming in front of, increasing funding for etc.) were included in several studies. There is no doubt that trained lifeguards are a reliable coastal drowning prevention strategy, but robust empirical evidence for their effectiveness is still scarce in the literature. The seminal report on lifeguard effectiveness was published by the United States Centers for Disease Control in 2001 and consisted primarily of expert opinion and a select number of case studies [81]. In the years since, research measuring the specific contributions by lifeguards to reducing or preventing drowning has made remarkably little progress [82]. Though evidence for lifeguard effectiveness from observational epidemiological studies or cost benefit analysis methods could make justifying existing or establishing new lifeguard services easier, lifeguards cannot be in all places at all times. It is therefore important to also invest in strategies which instil and motivate safe behaviour and practices around water among coastal users.

Signage was also a frequently recommended prevention strategy. Despite some water safety sign standards [83], uniformity is a persistent challenge and evidence exploring impact on behaviour suggests effectiveness is limited [57,84]. Moreover, traditional static signs are expensive (California State Parks estimated it would cost over $2M USD to add and update signs throughout the system) [85] and some land managers are now developing and implementing various forms of digital safety signs on beaches [86]. A number of important research questions related to coastal safety signage remain [87]; better evidence is needed to support an expensive strategy that could serve both a public liability and a public health/injury prevention purpose.

The dearth of prevention strategy evaluation is a major gap in the field that must be addressed by coastal drowning practitioners and researchers seeking to advance science that saves lives and prevents injury. Lack of evaluation of educational and other coastal safety programmes is not necessarily surprising [71], as confusion about the nature of program evaluation and limited funding for monitoring and evaluation has resulted in poorly conducted assessment in other injury areas [88]. Evidence that prevention programs and strategies work, or importantly, that they do not, is critical not only for strategic use of limited resources, but also for the design of future effective programs and initiatives. Evaluation research is most impactful when applied in contexts where it is needed most, which highlights another

significant challenge. Established coastal safety organisations in high income countries are probably best positioned in terms of resources and personnel to monitor, evaluate and share their work, but will their findings also be relevant for lower resourced settings? Are lessons transferable between settings with different beach and coastal hazards, safety and regulatory environments, economic realities, or cultural relationships with the water? Prioritising evaluation culture in the organisations that have the means, as well as bridging the gap to communities that don't, will be critical for continued progress in coastal drowning prevention and safety efforts.

## Limitations

Coastal drowning is an expansive area of research. This is the first study to attempt a broad review of the scope and nature of the field's published literature and should be considered with some limitations. First, it is possible that we did not capture every study meeting our inclusion criteria, although several steps including extensive manual searches were taken to ensure as broad a search as possible. While the results were charted and presented according to a framework rooted in epidemiological burden, risk factors and prevention strategies, the sheer volume and variety of information in these studies means that some coastal drowning topics included in the studies reviewed were not reported or discussed in this manuscript. Future reviews with specific aims and questions, versus a broad scoping purpose, may be better suited to evaluate these topics.

Importantly, the numerical summaries presented in this scoping review are not a substitute for meta-analysis methods which appraise and synthesize evidence to produce a clinically meaningful result. The results from this review reflect the state of coastal drowning research and do not account for the fact that many studies included were case series from specific datasets or geographic regions. Several risk factor reports from different studies conflicted, and in many areas a clear consensus was lacking. Future analysis addressing a specific question of this nature may offer new insight in these subjects, but a critical lesson is the importance of local data to drive prevention efforts.

Results of any one study should be taken with caution if they are to be applied to another geographic region. Similarly, the results from this review were dominated by studies from Australia and the United States and are likely a reflection of the places where research has been conducted, and the interests of the individual and groups or researchers involved, versus any sort of global consensus or understanding. Additionally, this review did not include epidemiological burden or risk factor information from non-fatal drowning cases, motivated in part by a desire to establish a more manageable review. Notwithstanding, we would expect to find similar results and derive comparable conclusions had we also included non-fatal information, although future research confirming such would be valuable.

## Conclusions

The coastal drowning literature is extensive and includes wide-ranging focus areas approached by multiple disciplines and various perspectives. This heterogeneity is both a challenge and a benefit. Standardisation in some areas would surely drive the science forward: consensus-based terminology and reporting practices would allow for richer data and improved comparisons across locations; a streamlined research agenda with identified objectives and prioritised questions to be addressed would propel prevention efforts towards the populations most in need; and established best practice for coastal drowning prevention program evaluation would ensure effective use of limited resources. Conversely, the disparate approach to coastal drowning research in the past 60 years has allowed for creative investigation of essential, ground-level questions that has pushed boundaries and driven the field in new directions.

The next generation of coastal drowning and safety science must build upon the advances of previous work by recognizing which areas have been studied thoroughly and where further attention is needed most. Coastal drowning researchers must address gaps in research from lower resourced settings and the lack of prevention strategy evaluation. The multidisciplinary nature of coastal drowning research offers collaborative opportunities to advance science underpinning prevention efforts seeking to save lives and keep people safe.

## Supporting information

**S1 Table. Included studies abstract table.**
(XLSX)

**S2 Table. Number of risk factor reports for each specific risk factor with related risk factor outcome by supporting data.**
(XLSX)

**S1 File. Preferred reporting items for systematic reviews and meta-analyses extension for scoping reviews (PRISMA-ScR) checklist.**
(PDF)

**S2 File. Coastal drowning scoping review search strategies.**
(PDF)

**S3 File. Coastal drowning scoping review data extraction form.**
(PDF)

## Acknowledgments

The authors thank Cheng Siu, UNSW Academic Services Librarian (Medicine), for her expert assistance with the database searches for this review, and Dr. Linda Quan, Harborview Injury Prevention and Research Center, for her input on the manuscript.

## Author Contributions

**Conceptualization:** William Koon, Amy Peden, Jasmin C. Lawes, Robert W. Brander.

**Data curation:** William Koon, Amy Peden.

**Formal analysis:** William Koon.

**Writing – original draft:** William Koon.

**Writing – review & editing:** William Koon, Amy Peden, Jasmin C. Lawes, Robert W. Brander.

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
