## [Decision Letter · Decision Letter 0]

1 Dec 2020

PONE-D-20-34091

Coastal drowning: A scoping review of burden, risk factors, and prevention strategies

PLOS ONE

Dear Dr. Koon,

Thank you for submitting your manuscript to PLOS ONE. After careful consideration, we feel that it has merit but does not fully meet PLOS ONE’s publication criteria as it currently stands. Therefore, we invite you to submit a revised version of the manuscript that addresses the points raised during the review process.

The methodology and results sections needs to be represented in a clear way. Kindly address this problem prior to moving forward with your manuscript.

Kindly address reviewer comments carefully since they are pivotal for improving the paper.

We look forward to receiving your revised manuscript.

Kind regards,

Sherief Ghozy, M.D., Ph.D. candidate

Academic Editor

PLOS ONE

Journal Requirements:

2. Please amend your list of authors on the manuscript to ensure that each author is linked to an affiliation. Authors’ affiliations should reflect the institution where the work was done (if authors moved subsequently, you can also list the new affiliation stating “current affiliation:….” as necessary).

Reviewers' comments:

Reviewer's Responses to Questions

**Comments to the Author**

1. Is the manuscript technically sound, and do the data support the conclusions?

Reviewer #1: Yes

Reviewer #2: Yes

2. Has the statistical analysis been performed appropriately and rigorously? 

Reviewer #1: Yes

Reviewer #2: N/A

3. Have the authors made all data underlying the findings in their manuscript fully available?

Reviewer #1: Yes

Reviewer #2: Yes

4. Is the manuscript presented in an intelligible fashion and written in standard English?

Reviewer #1: Yes

Reviewer #2: Yes

5. Review Comments to the Author

Reviewer #1: Coastal drowning: A scoping review of burden, risk factors, and prevention strategies

Manuscript Number: PONE-D-20-34091

Article Type: Research Article

Line 264-267:

Could you specify high income, upper-middle income, lower-middle income, and low-income countries?

Line 273-275:

The summation of 62 studies, 32, and 18 studies is not equal to 146.

Reviewer #2: Search strategy seems appropriate and comprehensive. However, the results reporting have significant gaps that make the results difficult to interpret. It appears that the authors have significantly underestimated the heterogeneity of the data.

6. PLOS authors have the option to publish the peer review history of their article (what does this mean?). If published, this will include your full peer review and any attached files.

Reviewer #1: No

Reviewer #2: No

---

## [Author Response · Author response to Decision Letter 0]

10 Jan 2021

Response to Reviewers 

Coastal drowning: A scoping review of burden, risk factors, and prevention strategies

PLOS ONE

Please see the attached file "response to reviewers" for a formatted version of this information, which may be easier to read. Thank you for your time.

Comments from Academic Editor

AE1 - Please ensure that your manuscript meets PLOS ONE's style requirements, including those for file naming. The PLOS ONE style templates can be found at

Thank you. We have reviewed the two links carefully and amended multiple components of the manuscript’s formatting including the title page, the references, supporting files/tables, and file naming.

AE2 - Please amend your list of authors on the manuscript to ensure that each author is linked to an affiliation. Authors’ affiliations should reflect the institution where the work was done (if authors moved subsequently, you can also list the new affiliation stating “current affiliation:….” as necessary).

We have reviewed the title page in relation to the PLOS One sample provided and updated so each author is linked to an affiliation in the required style.

AE3 - Please include captions for your Supporting Information files at the end of your manuscript, and update any in-text citations to match accordingly. Please see our Supporting Information guidelines for more information: http://journals.plos.org/plosone/s/supporting-information

Thank you, we have reviewed the PLOS One supporting information page and updated the manuscript to include the relevant supporting information caption details at the end of the manuscript. (Line 1138)

Comments from Reviewer #1

R1.1 - Line 264-267: Could you specify high income, upper-middle income, lower-middle income, and low-income countries?

Thank you for the opportunity to clarify this point. We classified each country according to the World Bank Country and Lending Groups for the year that each study was published. We cited the dataset (Reference 49) which is publicly available here and provides details on how the World Bank groups counties together and what thresholds were established for different categories in different years: https://datahelpdesk.worldbank.org/knowledgebase/articles/906519-world-bankcountry-and-lending-groups

We amended this sentence to read:

“Classified according to World Bank Country Income and Lending Groups at the time of publication (or nearest year from available data), 112 studies (76.7%) were from high income countries, 21 studies (14.4%) came from upper middle-income countries, 9 studies (6.2%) were from lower middle-income countries, and no studies were from low-income countries [49].”

Reference # 49:

The World Bank. Country and lending groups: historical classification by income. Washington D.C.: The World Bank, 2020. Available from: https://datahelpdesk.worldbank.org/knowledgebase/articles/906519-world-bank-country-and-lending-groups

R1.2 - Line 273-275: The summation of 62 studies, 32, and 18 studies is not equal to 146.

Thank you for bringing this sentence’s lack of clarity to our attention. We originally intended only to share the top three categories as not to add a long and excessively verbose sentence that provides duplicative information to that presented in Table 2, but realise how this may confuse the reader. We have clarified this point by amending the sentence to read:

“This cohort of coastal drowning research was published in 66 different journals; the three most frequently represented disciplines were non-injury-specific health/medical sciences (62 studies, 42.5%), followed by the physical sciences (32 studies, 21.9%), and those which were injury- or safety-specific (18 studies, 12.3%; Table 2).” 

Comments from Reviewer #2

R2.1 Search strategy seems appropriate and comprehensive. 

Thank you, we closely followed the detailed recommendations set forth by Arksey and O’Malley and Levec et al. 

R2.2 However, the results reporting have significant gaps that make the results difficult to interpret. 

Our original intent with this work was to “identify key concepts and factors studied, describing how they have been analysed and discussed, in order to synthesize present understanding and highlight gaps in the field.” To this end, we chose to conduct a broad scoping review in order to characterise the field as a whole so we might help future researchers answer more meaningful questions that will ultimately save more lives. In a topic as broad as coastal drowning, our author team had to make decisions about what to focus on and report in this manuscript and understood from the beginning that we would not be able to cover everything in this article alone. We therefore chose to report, in addition to characteristics of the studies themselves, information relevant to a public health framework including the epidemiological burden, risk factors and prevention strategies. 

We realise that there are a variety of other important topics in the coastal drowning literature that were not included in this manuscript. We have included supplementary files (see Tables S4 and S5) which provide more detail than we could include in the manuscript itself. We have re-written the limitations section to this end (see below, line 802-840 in the manuscript), including specific language noting that not every coastal drowning topic included in these papers was reported and discussed, and that future reviews with specific and refined questions vs a broad scoping purpose may be better suited to address some of these issues. Given these limitations, we reported results systematically using the framework outlined in the methods section: study characteristics, epidemiology, risk factors and prevention strategies, and also used several sub-headings to provide structure to the results reporting. 

The first paragraph of the limitations section now reads:

“Coastal drowning is an expansive area of research. This is the first study to attempt a broad review of the scope and nature of the field’s published literature and should be considered with some limitations. First, it is possible that we did not capture every study meeting our inclusion criteria, although several steps including extensive manual searches were taken to ensure as broad a search as possible. While the results were charted and presented according to a framework rooted in epidemiological burden, risk factors and prevention strategies, the sheer volume and variety of information in these studies means that some coastal drowning topics included in the studies reviewed were not reported or discussed in this manuscript. Future reviews with specific aims and questions, versus a broad scoping purpose, may be better suited to evaluate these topics.”

R2.4 It appears that the authors have significantly underestimated the heterogeneity of the data.

The heterogeneity of this body of coastal drowning research is in itself a major finding of this review. While we attempted to discuss this in several places in the discussion section (lines 585-9, 613-4, 617, 677, 680, 692-3, 809, 840-41), the reviewer comment has helped us realise we needed to address this in a more direct fashion, thank you for raising this point. We have therefore re-written the opening paragraph of the discussion and the conclusion section (see below) to acknowledge the diversity of information within these papers and consider how this has impacted the field. Thank you for the opportunity to clarify this. 

The opening paragraph of the discussion section now reads:

“This scoping review of 146 coastal drowning studies provides insight into the collective understanding of the coastal drowning problem and the science underpinning efforts to progress safety in this area. Our goal was to identify key concepts and factors in the peer reviewed coastal drowning literature and while some points of synthesis were identified, this review primarily shows the vast heterogeneity of science in the field. The factors and topics studied, how they have been investigated and what new knowledge they contributed are wide ranging and diverse. The populations included, data sources employed, and terminology used varied widely. A myriad of risk factors for coastal drowning and other preceding/proximal outcomes have been identified through analysis of different types of data using various methods. Recommended prevention strategies most frequently related to education, were commonly based on survey data or expert opinion alone and were rarely evaluated. Of note, the field’s scientific agenda to date has been, and continues to be, dominated by research from a small number of high-income countries.”

The conclusion now reads:

“The coastal drowning literature is extensive and includes wide-ranging focus areas approached by multiple disciplines and various perspectives. This heterogeneity is both a challenge and a benefit. Standardisation in some areas would surely drive the science forward: consensus-based terminology and reporting practices would allow for richer data and improved comparisons across locations; a streamlined research agenda with identified objectives and prioritised questions to be addressed would propel prevention efforts towards the populations most in need; and established best practice for coastal drowning prevention program evaluation would ensure effective use of limited resources. Conversely, the disparate approach to coastal drowning research in the past 60 years has allowed for creative investigation of essential, ground-level questions that has pushed boundaries and driven the field in new directions.

The next generation of coastal drowning and safety science must build upon the advances of previous work by recognizing which areas have been studied thoroughly and where further attention is needed most. Coastal drowning researchers must address gaps in research from lower resourced settings and the lack of prevention strategy evaluation. The multidisciplinary nature of coastal drowning research offers collaborative opportunities to advance science underpinning prevention efforts seeking to save lives and keep people safe.”

---

## [Editor Report · Decision Letter 1]

13 Jan 2021

Coastal drowning: A scoping review of burden, risk factors, and prevention strategies

PONE-D-20-34091R1

Dear Dr. Koon,

We’re pleased to inform you that your manuscript has been judged scientifically suitable for publication and will be formally accepted for publication once it meets all outstanding technical requirements.

Kind regards,

Sherief Ghozy, M.D., Ph.D. candidate

Academic Editor

PLOS ONE
---

## [Editor Report · Acceptance letter]

15 Jan 2021

PONE-D-20-34091R1 

Coastal drowning: A scoping review of burden, risk factors, and prevention strategies 

Dear Dr. Koon:

I'm pleased to inform you that your manuscript has been deemed suitable for publication in PLOS ONE. Congratulations! Your manuscript is now with our production department. 

Kind regards, 

on behalf of

Dr. Sherief Ghozy 

Academic Editor

PLOS ONE